# Training Restricted Boltzmann Machines via the Thouless-Anderson-Palmer Free Energy

**Marylou Gabrié**    **Eric W. Tramel**    **Florent Krzakala**
Laboratoire de Physique Statistique, UMR 8550 CNRS
École Normale Supérieure & Université Pierre et Marie Curie
75005 Paris, France
`{marylou.gabrie, eric.tramel}@lps.ens.fr, florent.krzakala@ens.fr`

## Abstract

Restricted Boltzmann machines are undirected neural networks which have been shown to be effective in many applications, including serving as initializations for training deep multi-layer neural networks. One of the main reasons for their success is the existence of efficient and practical stochastic algorithms, such as contrastive divergence, for unsupervised training. We propose an alternative deterministic iterative procedure based on an improved mean field method from statistical physics known as the Thouless-Anderson-Palmer approach. We demonstrate that our algorithm provides performance equal to, and sometimes superior to, persistent contrastive divergence, while also providing a clear and easy to evaluate objective function. We believe that this strategy can be easily generalized to other models as well as to more accurate higher-order approximations, paving the way for systematic improvements in training Boltzmann machines with hidden units.

## 1    Introduction

A restricted Boltzmann machine (RBM) [1, 2] is a type of undirected neural network with surprisingly many applications. This model has been used in problems as diverse as dimensionality reduction [3], classification [4], collaborative filtering [5], feature learning [6], and topic modeling [7]. Also, quite remarkably, it has been shown that generative RBMs can be stacked into multi-layer neural networks, forming an initialization for deep network architectures [8, 9]. Such deep architectures are believed to be crucial for learning high-order representations and concepts. Although the amount of training data available in practice has made pretraining of deep nets dispensable for supervised tasks, RBMs remain at the core of unsupervised learning, a key area for future developments in machine intelligence [10].

While the training procedure for RBMs can be written as a log-likelihood maximization, an exact implementation of this approach is computationally intractable for all but the smallest models. However, fast stochastic Monte Carlo methods, specifically contrastive divergence (CD) [2] and persistent CD (PCD) [11, 12], have made large-scale RBM training both practical and efficient. These methods have popularized RBMs even though it is not entirely clear why such approximate methods should work as well as they do.

In this paper, we propose an alternative deterministic strategy for training RBMs, and neural networks with hidden units in general, based on the so-called *mean-field*, and *extended mean-field*, methods of statistical mechanics. This strategy has been used to train neural networks in a number of earlier works [13, 14, 15, 16, 17]. In fact, for entirely visible networks, the use of adaptive cluster expansion mean-field methods has lead to spectacular results in learning Boltzmann machine representations [18, 19].

However, unlike these fully visible models, the hidden units of the RBM must be taken into account during the training procedure. In 2002, Welling and Hinton [17] presented a similar deterministic mean-field learning algorithm for general Boltzmann machines with hidden units, considering it *a priori* as a potentially efficient extension of CD. In 2008, Tieleman [12] tested the method in detail for RBMs and found it provided poor performance when compared to both CD and PCD. In the wake of these two papers, little inquiry has been made in this direction, with the apparent consensus being that the deterministic mean-field approach is ineffective for RBM training.

Our goal is to challenge this consensus by going beyond naïve mean field, a mere first-order approximation, by introducing second-, and possibly third-, order terms. In principle, it is even possible to extend the approach to arbitrary order. Using this extended mean-field approximation, commonly known as the Thouless-Anderson-Palmer [20] approach in statistical physics, we find that RBM training performance is significantly improved over the naïve mean-field approximation and is even comparable to PCD. The clear and easy to evaluate objective function, along with the extensible nature of the approximation, paves the way for systematic improvements in learning efficiency.

## 2 Training restricted Boltzmann machines

A restricted Boltzmann machine, which can be viewed as a two layer undirected bipartite neural network, is a specific case of an energy based model wherein a layer of visible units is fully connected to a layer of hidden units. Let us denote the binary visible and hidden units, indexed by $i$ and $j$ respectively, as $v_i$ and $h_j$. The energy of a given state, $\mathbf{v} = \{v_i\}$, $\mathbf{h} = \{h_j\}$, of the RBM is given by

$$E(\mathbf{v}, \mathbf{h}) = -\sum_i a_i v_i - \sum_j b_j h_j - \sum_{i,j} v_i W_{ij} h_j, \tag{1}$$

where $W_{ij}$ are the entries of the matrix specifying the weights, or *couplings*, between the visible and hidden units, and $a_i$ and $b_j$ are the biases, or the *external fields* in the language of statistical physics, of the visible and hidden units, respectively. Thus, the set of parameters $\{W_{ij}, a_i, b_j\}$ defines the RBM model.

The joint probability distribution over the visible and hidden units is given by the Gibbs-Boltzmann measure $P(\mathbf{v}, \mathbf{h}) = Z^{-1} e^{-E(\mathbf{v}, \mathbf{h})}$, where $Z = \sum_{\mathbf{v}, \mathbf{h}} e^{-E(\mathbf{v}, \mathbf{h})}$ is the normalization constant known as the *partition function* in physics. For a given data point, represented by $\mathbf{v}$, the marginal of the RBM is calculated as $P(\mathbf{v}) = \sum_{\mathbf{h}} P(\mathbf{v}, \mathbf{h})$. Writing this marginal of $\mathbf{v}$ in terms of its log-likelihood results in the difference

$$\mathcal{L} = \ln P(\mathbf{v}) = -F^c(\mathbf{v}) + F, \tag{2}$$

where $F = -\ln Z$ is the *free energy* of the RBM, and $F^c(\mathbf{v}) = -\ln(\sum_{\mathbf{h}} e^{-E(\mathbf{v}, \mathbf{h})})$ can be interpreted as a free energy as well, but with visible units fixed to the training data point $\mathbf{v}$. Hence, $F^c$ is referred to as the *clamped* free energy.

One of the most important features of the RBM model is that $F^c$ can be easily computed as $\mathbf{h}$ may be summed out analytically since the hidden units are conditionally independent of the visible units, owing to the RBM's bipartite structure. However, calculating $F$ is computationally intractable since the number of possible states to sum over scales combinatorially with the number of units in the model. This complexity frustrates the exact computation of the gradients of the log-likelihood needed in order to train the RBM parameters via gradient ascent. Monte Carlo methods for RBM training rely on the observation that $\frac{\partial F}{\partial W_{ij}} = P(v_i = 1, h_j = 1)$, which can be simulated at a lower computational cost. Nevertheless, drawing independent samples from the model in order to approximate this derivative is itself computationally expensive and often approximate sampling algorithms, such as CD or PCD, are used instead.

## 3 Extended mean field theory of RBMs

Here, we present a physics-inspired tractable estimation of the free energy $F$ of the RBM. This approximation is based on a high temperature expansion of the free energy derived by Georges and Yedidia in the context of spin glasses [21] following the pioneering works of [20, 22]. We refer the reader to [23] for a review of this topic.

To apply the Georges-Yedidia expansion to the RBM free energy, we start with a general energy based model which possesses arbitrary couplings $W_{ij}$ between undifferentiated binary spins $s_i \in \{0, 1\}$, such that the energy of the Gibbs-Boltzmann measure on the configuration $\mathbf{s} = \{s_i\}$ is defined by $E(\mathbf{s}) = -\sum_i a_i s_i - \sum_{(i,j)} W_{ij} s_i s_j$[1]. We also restore the role of the temperature, usually considered constant and for simplicity set to 1 in most energy based models, by multiplying the energy functional in the Boltzmann weight by the inverse temperature $\beta$.

Next, we apply a Legendre transform to the free energy, a standard procedure in statistical physics, by first writing the free energy as a function of a newly introduced auxiliary external field $\mathbf{q} = \{q_i\}$, $-\beta F[\mathbf{q}] = \ln \sum_{\mathbf{s}} e^{-\beta E(\mathbf{s}) + \beta \sum_i q_i s_i}$. This external field will be eventually set to the value $\mathbf{q} = \mathbf{0}$ in order to recover the true free energy. The Legendre transform $\Gamma$ is then given as a function of the conjugate variable $\mathbf{m} = \{m_i\}$ by maximizing over $\mathbf{q}$,

$$-\beta \Gamma[\mathbf{m}] = -\beta \max_{\mathbf{q}} [F[\mathbf{q}] + \sum_i q_i m_i] = -\beta (F[\mathbf{q}^*[\mathbf{m}]] + \sum_i q_i^*[\mathbf{m}] m_i), \quad (3)$$

where the maximizing auxiliary field $\mathbf{q}^*[\mathbf{m}]$, a function of the conjugate variables, is the inverse function of $\mathbf{m}[\mathbf{q}] \equiv -\frac{dF}{d\mathbf{q}}$. Since the derivative $\frac{dF}{d\mathbf{q}}$ is exactly equal to $-\langle \mathbf{s} \rangle$, where the operator $\langle \cdot \rangle$ refers to the average configuration under the Boltzmann measure, the conjugate variable $\mathbf{m}$ is in fact the equilibrium *magnetization* vector $\langle \mathbf{s} \rangle$. Finally, we observe that the free energy is also the *inverse* Lengendre transform of its Legendre transform at $\mathbf{q} = \mathbf{0}$,

$$-\beta F = -\beta F[\mathbf{q} = \mathbf{0}] = \beta \min_{\mathbf{m}} [\Gamma[\mathbf{m}]] = -\beta \Gamma[\mathbf{m}^*], \quad (4)$$

where $\mathbf{m}^*$ minimizes $\Gamma$, which yields an expression of the free energy in terms of the magnetization vector. Following [22, 21], this formulation allows us to perform a high temperature expansion of $A(\beta, \mathbf{m}) \equiv -\beta \Gamma[\mathbf{m}]$ around $\beta = 0$ at fixed $\mathbf{m}$,

$$A(\beta, \mathbf{m}) = A(0, \mathbf{m}) + \beta \left. \frac{\partial A(\beta, \mathbf{m})}{\partial \beta} \right|_{\beta=0} + \frac{\beta^2}{2} \left. \frac{\partial^2 A(\beta, \mathbf{m})}{\partial \beta^2} \right|_{\beta=0} + \cdots, \quad (5)$$

where the dependence on $\beta$ of the product $\beta \mathbf{q}$ must carefully be taken into account. At infinite temperature, $\beta = 0$, the spins decorrelate, causing the average value of an arbitrary product of spins to equal the product of their local magnetizations; a useful property. Accounting for binary spins taking values in $\{0, 1\}$, one obtains the following expansion

$$-\beta \Gamma(\mathbf{m}) = -\sum_i [m_i \ln m_i + (1 - m_i) \ln(1 - m_i)] + \beta \sum_i a_i m_i + \beta \sum_{(i,j)} W_{ij} m_i m_j$$

$$+ \frac{\beta^2}{2} \sum_{(i,j)} W_{ij}^2 (m_i - m_i^2)(m_j - m_j^2)$$

$$+ \frac{2\beta^3}{3} \sum_{(i,j)} W_{ij}^3 (m_i - m_i^2) \left( \frac{1}{2} - m_i \right) (m_j - m_j^2) \left( \frac{1}{2} - m_j \right)$$

$$+ \beta^3 \sum_{(i,j,k)} W_{ij} W_{jk} W_{ki} (m_i - m_i^2)(m_j - m_j^2)(m_k - m_k^2) + \cdots \text{[1]} \quad (6)$$

The zeroth-order term corresponds to the entropy of non-interacting spins with constrained magnetizations values. Taking this expansion up to the first-order term, we recover the standard naïve mean-field theory. The second-order term is known as the *Onsager reaction term* in the TAP equations [20]. The higher orders terms are systematic corrections which were first derived in [21].

Returning to the RBM notation and truncating the expansion at second-order for the remainder of the theoretical discussion, we have

$$\Gamma(\mathbf{m}^v, \mathbf{m}^h) \approx S(\mathbf{m}^v, \mathbf{m}^h) - \sum_i a_i m_i^v - \sum_j b_j m_j^h$$

$$- \sum_{i,j} W_{ij} m_i^v m_j^h + \frac{W_{ij}^2}{2} (m_i^v - (m_i^v)^2)(m_j^h - (m_j^h)^2), \quad (7)$$

where $S$ is the entropy contribution, $\mathbf{m^v}$ and $\mathbf{m^h}$ are introduced to denote the magnetization of the visible and hidden units, and $\beta$ is set equal to 1. Eq. (7) can be viewed as a *weak coupling* expansion in $W_{ij}$. To recover an estimate of the RBM free energy, Eq. (7) must be minimized with respect to its arguments, as in Eq. (4). Lastly, by writing the stationary condition $\frac{d\Gamma}{d\mathbf{m}} = \mathbf{0}$, we obtain the self-consistency constraints on the magnetizations. At second-order we obtain the following constraint on the visible magnetizations,

$$m_i^v \approx \text{sigm} \left[ a_i + \sum_j W_{ij} m_j^h - W_{ij}^2 \left( m_i^v - \frac{1}{2} \right) \left( m_j^h - (m_j^h)^2 \right) \right], \tag{8}$$

where $\text{sigm}[x] = (1 + e^{-x})^{-1}$ is a logistic sigmoid function. A similar constraint must be satisfied for the hidden units, as well. Clearly, the stationarity condition for $\Gamma$ obtained at order $n$ utilizes terms up to the $n^{th}$ order within the sigmoid argument of these consistency relations. Whatever the order of the approximation, the magnetizations are the solutions of a set of non-linear coupled equations of the same cardinality as the number of units in the model. Finally, provided we can define a procedure to efficiently derive the value of the magnetizations satisfying these constraints, we obtain an extended mean-field approximation of the free energy which we denote as $F^{\text{EMF}}$.

## 4 RBM evaluation and unsupervised training with EMF

### 4.1 An iteration for calculating $F^{\text{EMF}}$

Recalling the log-likelihood of the RBM, $\mathcal{L} = -F^c(\mathbf{v}) + F$, we have shown that a tractable approximation of $F$, $F^{\text{EMF}}$, is obtained via a weak coupling expansion so long as one can solve the coupled system of equations over the magnetizations shown in Eq. (8). In the spirit of iterative belief propagation [23], we propose that these self-consistency relations can serve as update rules for the magnetizations within an iterative algorithm. In fact, the convergence of this procedure has been rigorously demonstrated in the context of random spin glasses [24]. We expect that these convergence properties will remain present even for real data. The iteration over the self-consistency relations for both the hidden and visible magnetizations can be written using the time index $t$ as

$$m_j^h[t+1] \leftarrow \text{sigm} \left[ b_j + \sum_i W_{ij} m_i^v[t] - W_{ij}^2 \left( m_j^h[t] - \frac{1}{2} \right) \left( m_i^v[t] - (m_i^v[t])^2 \right) \right], \tag{9, 10}$$

$$m_i^v[t+1] \leftarrow \text{sigm} \left[ a_i + \sum_j W_{ij} m_j^h[t+1] - W_{ij}^2 \left( m_i^v[t] - \frac{1}{2} \right) \left( m_j^h[t+1] - (m_j^h[t+1])^2 \right) \right],$$

where the time indexing follows from application of [24]. The values of $\mathbf{m}^v$ and $\mathbf{m}^h$ minimizing $\Gamma(\mathbf{m}^v, \mathbf{m}^h)$, and thus providing the value of $F^{\text{EMF}}$, are obtained by running Eqs. (9, 10) until they converge to a fixed point. We note that while we present an iteration to find $F^{\text{EMF}}$ up to second-order above, third-order terms can easily be introduced into the procedure.

### 4.2 Deterministic EMF training

By using the EMF estimation of $F$, and the iterative algorithm detailed in the previous section to calculate it, it is now possible to estimate the gradients of the log-likelihood used for unsupervised training of the RBM model by substituting $F$ with $F^{\text{EMF}}$. We note that the deterministic iteration we propose for estimating $F$ is in stark contrast with the stochastic sampling procedures utilized in CD and PCD to the same end. The gradient ascent update of weight $W_{ij}$ is approximated as

$$\Delta W_{ij} \propto \frac{\partial \mathcal{L}}{\partial W_{ij}} \approx -\frac{\partial F^c}{\partial W_{ij}} + \frac{\partial F^{\text{EMF}}}{\partial W_{ij}}, \tag{11}$$

where $\frac{\partial F^{\text{EMF}}}{\partial W_{ij}}$ can be computed by differentiating Eq. (7) at fixed $\mathbf{m}^v$ and $\mathbf{m}^h$ and computing the value of this derivative at the fixed points of Eqs. (9, 10) obtained from the iterative procedure. The gradients with respect to the visible and hidden biases can be derived similarly. Interestingly, $\frac{\partial F^{\text{EMF}}}{\partial a_i}$

and $\frac{\partial F^{\text{EMF}}}{\partial b_j}$ are merely the fixed-point magnetizations of the visible and hidden units, $m_i^v$ and $m_j^h$, respectively.

*A priori*, the training procedure sketched above can be used at any order of the weak coupling expansion. The training algorithm introduced in [17], which was shown to perform poorly for RBM training in [12], can be recovered by retaining only the first-order of the expansion when calculating $F^{\text{EMF}}$. Taking $F^{\text{EMF}}$ to second-order, we expect that training efficiency and performance will be greatly improved over [17]. In fact, including the third-order term in the training algorithm is just as easy as including the second-order one, due to the fact that the particular structure of the RBM model does not admit triangles in its corresponding factor graphs. Although the third-order term in Eq. (6) does include a sum over distinct pairs of units, as well as a sum over coupled triplets of units, such triplets are excluded by the bipartite structure of the RBM. However, coupled *quadruplets* do contribute to the fourth-order term and therefore fourth- and higher-order approximations require much more expensive computations [21], though it is possible to utilize adaptive procedures [19].

## 5 Numerical experiments

### 5.1 Experimental framework

To evaluate the performance of the proposed deterministic EMF RBM training algorithm[1], we perform a number of numerical experiments over two separate datasets and compare these results with both CD-1 and PCD. We first use the MNIST dataset of labeled handwritten digit images [25]. The dataset is split between $60\,000$ training images and $10\,000$ test images. Both subsets contain approximately the same fraction of the ten digit classes (0 to 9). Each image is comprised of $28 \times 28$ pixels taking values in the range $[0, 255]$. The MNIST dataset was binarized by setting all non-zero pixels to 1 in all experiments.

Second, we use the $28 \times 28$ pixel version of the Caltech 101 Silhouette dataset [26]. Constructed from the Caltech 101 image dataset, the silhouette dataset consists of black regions of the primary foreground scene objects on a white background. The images are labeled according to the object in the original picture, of which there are 101 unevenly represented object labels. The dataset is split between a training ($4\,100$ images), a validation ($2\,264$ images), and a test ($2\,304$ images) sets.

For both datasets, the RBM models require 784 visible units. Following previous studies evaluating RBMs on these datasets, we fix the number of RBM hidden units to 500 in all our experiments. During training, we adopt the mini-batch learning procedure for gradient averaging, with 100 training points per batch for MNIST and 256 training points per batch for Caltech 101 Silhouette.

We test the EMF learning algorithm presented in Section 4.2 in various settings. First, we compare implementations utilizing the first-order (MF), second-order (TAP2), and third-order (TAP3) approximations of $F$. Higher orders were not considered due to their greater complexity. Next, we investigate training quality when the self-consistency relations on the magnetizations were not converged when calculating the derivatives of $F^{\text{EMF}}$, instead iterated for a small, fixed (3) number of times, an approach similar to CD. Furthermore, we also evaluate a "persistent" version of our algorithm, similar to [12]. As in PCD, the iterative EMF procedure possesses multiple initialization-dependent fixed-point magnetizations. Converging multiple chains allows us to collect proper statistics on these basins of attraction. In this implementation, the magnetizations of a set of points, dubbed fantasy particles, are updated and maintained throughout the training in order to estimate $F$. This persistent procedure takes advantage of the fact that the RBM-defined Boltzmann measure changes only slightly between parameter updates. Convergence to the new fixed point magnetizations at each minibatch should therefore be sped up by initializing with the converged state from the previous update. Our final experiments consist of persistent training algorithms using 3 iterations of the magnetization self-consistency relations (P-MF, P-TAP2 and P-TAP3) and one persistent training algorithm using 30 iterations (P-TAP2-30) for comparison.

For comparison, we also train RBM models using CD-1, following the prescriptions of [27], and PCD, as implemented in [12]. Given that our goal is to compare RBM training approaches rather than achieving the best *possible* training across all free parameters, neither momentum nor adaptive learning rates were included in any of the implementations tested. However, we do employ a weight

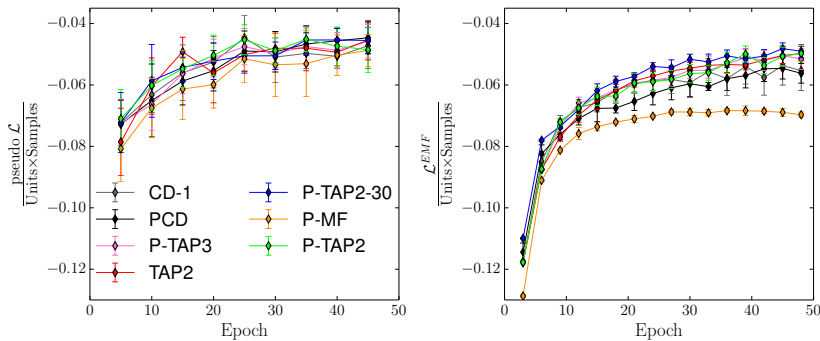

Figure 1: Estimates of the per-sample log-likelihood over the MNIST test set, normalized by the total number of units, as a function of the number of training epochs. The results for the different training algorithms are plotted in different colors with the same color code used for both panels. **Left panel :** Pseudo log-likelihood estimate. The difference between EMF algorithms and contrastive divergence algorithms is minimal. **Right panel :** EMF log-likelihood estimate at $2^{nd}$ order. The improvement from MF to TAP is clear. Perhaps reasonably, TAP demonstrates an advantage over CD and PCD. Notice how the second-order EMF approximation of $\mathcal{L}$ provides less noisy estimates, at a lower computational cost.

decay regularization in all our trainings to keep weights small; a necessity for the weak coupling expansion on which the EMF relies. When comparing learning procedures on the same plot, all free parameters of the training (e.g. learning rate, weight decay, etc.) were set identically. All results are presented as averages over 10 independent trainings with standard deviations reported as error bars.

## 5.2 Relevance of the EMF log-likelihood

Our first observation is that the implementations of the EMF training algorithms are not overly belabored. The free parameters relevant for the PCD and CD-1 procedures were found to be equally well suited for the EMF training algorithms. In fact, as shown in the left panel of Fig. 1, and the right inset of Fig. 3, the ascent of the pseudo log-likelihood over training epochs is very similar between the EMF training methods and both the CD-1 and PCD trainings.

Interestingly, for the Caltech 101 Silhouettes dataset, it seems that the persistent algorithms tested have difficulties in ascending the pseudo-likelihood in the first epochs of training. This contradicts the common belief that persistence yields more accurate approximations of the likelihood gradients. The complexity of the training set, 101 classes unevenly represented over only $4\,100$ training points, might explain this unexpected behavior. The persistent fantasy particles all converge to similar non-informative blurs in the earliest training epochs with many epochs being required to resolve the particles to a distribution of values which are informative about the pseudo log-likelihood.

Examining the fantasy particles also gives an idea of the performance of the RBM as a generative model. In Fig. 2, 24 randomly chosen fantasy particles from the $50^{th}$ epoch of training with PCD, P-MF, and P-TAP2 are displayed. The RBM trained with PCD generates recognizable digits, yet the model seems to have trouble generating several digit classes, such as 3, 8, and 9. The fantasy particles extracted from a P-MF training are of poorer quality, with half of the drawn particles featuring non-identifiable digits. The P-TAP2 algorithm, however, appears to provide qualitative improvements. All digits can be visually discerned, with visible defects found only in two of the particles. These particles seem to indicate that it is indeed possible to efficiently persistently train an RBM without converging on the fixed point of the magnetizations.

The relevance of the EMF log-likelihood for RBM training is further confirmed in the right panel of Fig. 1, where we observe that both CD-1 and PCD ascend the second-order EMF log-likelihood, *even though they are not explicitly constructed to optimize over this objective*. As expected, the persistent TAP2 algorithm with 30 iterations of the magnetizations (P-TAP2-30) achieves the best maximization of $\mathcal{L}^{EMF}$. However, P-TAP2, with only 3 iterations of the magnetizations, achieves very similar performance, perhaps making it preferable when a faster training algorithm is desired.

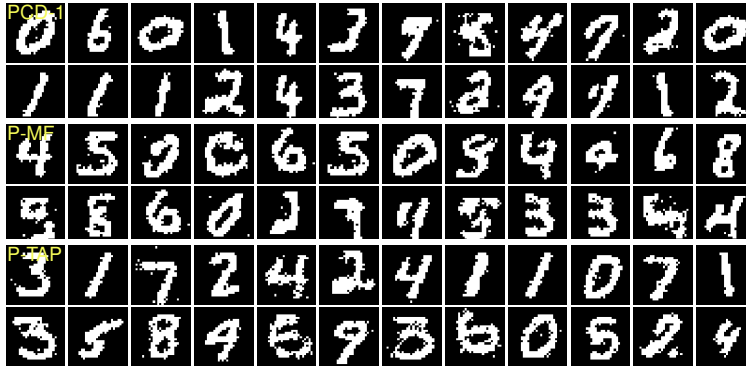

Figure 2: Fantasy particles generated by a 500 hidden unit RBM after 50 epochs of training on the MNIST dataset with PCD (**top two rows**), P-MF (**middle two rows**) and P-TAP2 (**bottom two rows**). These fantasy particles represent typical samples generated by the trained RBM when used as a generative prior for handwritten numbers. The samples generated by P-TAP2 are of similar subjective quality, and perhaps slightly preferable, to those generated by PCD, while certainly preferable to those generated by P-MF.

Moreover, we note that although P-TAP2 demonstrates improvements with respect to the P-MF, the P-TAP3 does not yield significantly better results than P-TAP2. This is perhaps not surprising since the third order term of the EMF expansion consists of a sum over as many terms as the second order, but at a smaller order in $\{W_{ij}\}$.

Lastly, we note the computation times for each of these approaches. For a Julia implementation of the tested RBM training techniques running on a 3.2 GHz Intel i5 processor, we report the 10 trial average wall times for fitting a single 100-sample batch normalized against the model complexity. PCD, which uses only a single sampling step, required $14.10\pm0.97$ $\mu$s/batch/unit. The three EMF techniques, P-MF, P-TAP2, and P-TAP3, each of which use 3 magnetization iterations, required $21.25 \pm 0.22$ $\mu$s/batch/unit, $37.22 \pm 0.34$ $\mu$s/batch/unit, and $64.88 \pm 0.45$ $\mu$s/batch/unit, respectively. If fewer magnetization iterations are required, as we have empirically observed in limited tests, then the run times of the P-MF and P-TAP2 approaches are commesurate with PCD.

### 5.3 Classification task performance

We also evaluate these RBM training algorithms from the perspective of supervised classification. An RBM can be interpreted as a deterministic function mapping the binary visible unit values to the real-valued hidden unit magnetizations. In this case, the hidden unit magnetizations represent the contributions of some learned features. Although no supervised fine-tuning of the weights is implemented, we tested the quality of the features learned by the different training algorithms by their usefulness in classification tasks. For both datasets, a logistic regression classifier was calibrated with the hidden units magnetizations mapped from the labeled training images using the `scikit-learn` toolbox [28]. We purposely avoid using more sophisticated classification algorithms in order to place emphasis on the quality of the RBM training, not the classification method.

In Fig. 3, we see that the MNIST classification accuracy of the RBMs trained with the P-TAP2 algorithms is roughly equivalent with that obtained when using PCD training, while CD-1 training yields markedly poorer classification accuracy. The slight decrease in performance of CD-1 and TAP2 along as the training epochs increase might be emblematic of over-fitting by the non-persistent algorithms, although no decrease in the EMF test set log-likelihood was observed.

Finally, for the Caltech 101 Silhouettes dataset, the classification task, shown in the right panel of Fig. 3, is much more difficult *a priori*. Interestingly, the persistent algorithms do not yield better results on this task. However, we observe that the performance of deterministic EMF RBM training is at least comparable with both CD-1 and PCD.

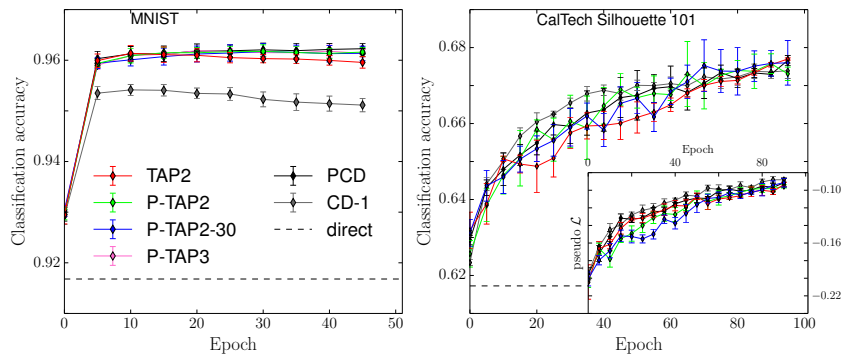

Figure 3: Test set classification accuracy for the MNIST (**left**) and Caltech 101 Silhouette (**right**) datasets using logistic regression on the hidden-layer marginal probabilities as a function of the number of epochs. As a baseline comparison, the classification accuracy of logistic regression performed directly on the data is given as a black dashed line. The results for the different training algorithms are displayed in different colors, with the same color code being used in both panels. (**Right inset:**) Pseudo log-likelihood over training epochs for the Caltech 101 Silhouette dataset.

## 6   Conclusion

We have presented a method for training RBMs based on an extended mean field approximation. Although a naïve mean field learning algorithm had already been designed for RBMs, and judged unsatisfactory [17, 12], we have shown that extending beyond the naïve mean field to include terms of second-order and above brings significant improvements over the first-order approach and allows for practical and efficient deterministic RBM training with performance comparable to the stochastic CD and PCD training algorithms.

The extended mean field theory also provides an estimate of the RBM log-likelihood which is easy to evaluate and thus enables practical monitoring of the progress of unsupervised learning throughout the training epochs. Furthermore, training on real-valued magnetizations is theoretically well-founded within the presented approach, paving the way for many possible extensions. For instance, it would be quite straightforward to apply the same kind of expansion to Gauss-Bernoulli RBMs, as well as to multi-label RBMs.

The extended mean field approach might also be used to learn stacked RBMs jointly, rather than separately, as is done in both deep Boltzmann machine and deep belief network pre-training, a strategy that has shown some promise [29]. In fact, the approach can be generalized even to non-restricted Boltzmann machines with hidden variables with very little difficulty. Another interesting possibility would be to make use of higher-order terms in the series expansion using adaptive cluster methods such as those used in [19]. We believe our results show that the extended mean field approach, and in particular the Thouless-Anderson-Palmer one, may be a good starting point to theoretically analyze the performance of RBMs and deep belief networks.

**Acknowledgments**

We would like to thank F. Caltagirone and A. Decelle for many insightful discussions. This research was funded by European Research Council under the European Union's 7[th] Framework Programme (FP/2007-2013/ERC Grant Agreement 307087-SPARCS).

## Footnotes

[1]The notation $\sum_{(i,j)}$ and $\sum_{(i,j,k)}$ refers to the sum over the distinct pairs and triplets of spins, respectively.

[1]Available as a Julia package at `https://github.com/sphinxteam/Boltzmann.jl`

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
