[Reviews · NeurIPS 2015]

Submitted by Assigned_Reviewer_1

Detailed comments: 40 - You might be interested in [MacKay, David. "Failures of the one-step learning algorithm." 2001.]

42 - PCD is a rediscovery of a method that has recurred several times before, and which I believe actually has some nice convergence properties, at least in terms of some limit in the ratio of learning rate and mixing time. [Neal, R. M. (1992). Connectionist learning of belief networks. Artificial Intelligence] is one earlier example of its use that I'm familiar with. A paper by Salakhutdinov suggests that [H. Robbins and S. Monro. A stochastic approximation method. Ann. Math. Stat., 22:400-407, 1951.] is the earliest reference, though I have not read this older paper.

48 - Though they tend to break down around criticality, which is exactly where many interesting real world datasets are expected to be. e.g. see mean field theory with TAP doing worse than the random parameter initialization in [Sohl-Dickstein, Jascha, Peter Battaglino, and Michael R. DeWeese. "Minimum probability flow learning." 2011.]

72 - nit - common convention is lowercase for vectors, upper case for matrices. maybe w_ij -> W_ij

97 - since you never have to sample the visible units, you could us the more flexible sRBMs, rather than RBMs (sRBM = semi-restricted Boltzmann machine, an RBM w/ the addition of visible to visible couplings) (updated -- I see later that when you take your expansion to third order, this would make the sRBM expensive)

132 - "where the dependence on beta of the product ..." huh? This didn't make sense to me.

194 - re: careful application, what makes this part tricky?

202 - "to gradients" -> "the gradients"

210 - i.e. implicit differentiation

254 - Why do you need multiple points? Doesn't m have a single fixed point?

262-267 - Confused by this paragraph.

Figure 1 - Label units on the y-axis! right pane - Should note that this pane is evaluating using the same approx. objective as used in training, so not a fair comparison for CD.

296 - re: all parameters of training being set identically - this is not fair! For instance, the training gradient in PCD/CD will have a higher variance, and so it will probably perform best w/ a smaller learning rate than EMF. Might consider using a tool like Spearmint [Snoek, Jasper, Hugo Larochelle, and Ryan P. Adams. "Practical Bayesian optimization of machine learning algorithms." Advances in neural information processing systems. 2012.] as a black box (and thus parsimonious) way to set hyperparameters.

Figure 2 - These look grayscale not binary. Are they actually samples or are they probabilities of activation? Make sure to label correctly. Samples give a more clear view of performance. - Curious to see P-TAP3!

371-377 - What is the computational cost of these algorithms? What is the wall clock time relative to PCD?

391 - "over-fitting" suggests that it's getting better on the training data. It's probably not.

(one more comment after reading review from Assigned_Reviewer_4 -- score matching would not apply to this specific case, with RBMs with binary visible units. There are a number of other parameter estimation techniques for energy based models that would however -- minimum probability flow, ratio matching, noise contrastive estimation, minimum KL contraction.)
Summary: Overall I liked the paper. It applies higher order mean field techniques to train RBMs. Better ways to train multi-layer probabilistic models are interesting, and address a major bottleneck in working with them. The experimental results convinced me that it works on simple datasets, but did not at all convince me it's better or faster than standard techniques. I also have a suspicion the algorithm would fail on many datasets -- e.g. those with power-law/multiscale/critical structure, as is found in many interesting ML datasets.

Submitted by Assigned_Reviewer_2

The authors present an extended mean field approximation for the free energy of a restricted Boltzmann machine, and provide experiments showing that it performs better than a previous mean-field approach.

I am unsure about the originality or significance of this paper in the context of research on RBMs/DBMs, but I found it to be clear and of high quality.

The experiments are well-done; I like how they include versions with different numbers of iterations in the TAP estimate and both the persistent and nonpersistent initializations.

Summary: I am unsure about the originality of this paper, but it was clear and enjoyable to read.

Submitted by Assigned_Reviewer_3

This paper proposes to use a deterministic iterative procedure for training RBMs. This procedure is based on an extended mean-field method which is effective for both generative and discriminative purposes.

+ the theory are explained with clarity.

+ although the theory and problem are both well-established, the marriage of the two is novel.

+ the experiments are well thought out and carried out.

- score matching, another deterministic training strategy has already been proposed and used to train energy based models, see Kingma & LeCun NiPS 2010, and Marlin, et al AISTATS 2010. This reduces the significance of the paper's contributions.

- there lacks an experiment or an argument that showcases the advantage of the proposed method. All we see is how TAP is a great approximation to PCDs in turns of the training outcome, but we do not see in what sense TAP is superior, i.e. what is an empirical demonstration of the advantage of having a deterministic objective function? Maybe having an objective enables second-order optimization methods and accelerates learning? Anyways one additional experiments on this direction would be nice.

- In the past RBMs are used primary for pre-training deep nets, but now this has become less relevant since deep nets can be trained using stochastic gradient descent over lots of data, hence research on RBMs halted (although I agree with the authors that it shouldn't). One sentence on why RBMs are still very relevant in the introduction would be helpful.

Summary: This paper proposes a deterministic objective function for RBMs which make them easy to train. While this paper has a strong theoretical aspect, it fails to demonstrate experimentally the advantage of the proposed method.

Submitted by Assigned_Reviewer_4

Paper 443 Training Restricted Boltzmann Machines via the Thouless-Anderson-Palmer Free Energy

This paper reconsiders the mean field method for estimating the parameters of a restricted Boltzmann machine. The main idea is to extend the analysis beyond the most common first order approximation and consider the Thouless-Andersion-Palmer approach.

The method offers an alternative to the common algorithms CD and PCD.

While previous work based on first order mean field approximations revealed poor performance,

the present paper demonstrates that the extended mean field method can achieve a performance comparable to PCD.

The paper is very well written. The motivation is clear. The execution is transparent.

COMMENTS:

The second paragraph of the introduction could cite some of the works that have investigated contrastive divergence.

It would be good to remind the reader at strategic points why $\beta$ is introduced/set to $1$.

``At infinite temperature, ... causing the average value of an arbitrary product of spins...''

This may be a matter of terminology, but as stated the claim holds irrespective of the temperature.

The symbol $\simeq$ should be explained or replaced by a less ambiguous symbol.

In eq. 7 and 8 it would be good to use parenthesis, as $(m_i^v)^2$.

The meaning of eq. 4 could be highlighted or, alternatively, at some point spend a few more words on the relation between $\Gamma$ and $F$.

After eq. 11 it would be good to be more explicit with ``obtained from the iterative procedure.''

Introduce the pseudo log-likelihood.

It would be interesting to read more about how 3rd order terms do not contribute as much to the performance of the method.

Summary: The paper is very well written. The motivation is clear, the execution is transparent, and the results are interesting.

Author Feedback
Author rebuttal: Dear NIPS committee,

We thank the reviewers for their valuable feedback and their overall appreciation of our work. We will make sure to correct all of the mentioned typos and ambiguities in the final version of our paper. Below, we detail our responses to the specific comments of the referees.

** Response to Reviewer 2:

(48): Interestingly, the method of [16] (which moves to higher and higher orders of the TAP extension) appears to work very well, empirically, even close to criticality. It does become slower around this point, however.

(97) : We have already begun to study the semi-RBM with TAP2 for a future publication.

(132): Here, $q$ is itself a function of $\beta$. In an attempt to keep the equations clear, we did not specify this in the notation.

(194): Though the time indices are natural in this bipartite context with visible and hidden layers, their derivation becomes more involved in the non-bipartite case (e.g. the semi-RBM). In the context of this paper, this remark is indeed misleading.

(254): Interestingly, much like a Hopfield network, there exist a large number of fixed-points in the energy landscape for $m$. Therefore, one may obtain a different solution depending on the initial conditions. Hence, multiple particles.

(262-267): We shall perhaps remove this paragraph, which is both unclear and too preliminary, and reserve such analysis for a forthcoming publication on continuous RBMs and stacked RBMs.

(Figure 2): Yes, these are the magnetizations. We could instead show binary samples.

(371-377): Depending on the implementation, we found TAP2 wall-time to be of the same order, but slightly slower, than PCD. We can include this comparison in the final paper.

(391): OK

** Response to Reviewer 3:

- We agree with the comments and shall clarify these points.

** Response to Reviewer 4:

- Score matching does not apply here as it relies on properties of continuous data (see Reviewer 3's comments). We agree that there are, however, a number of different techniques that can be used in practice. We choose to compare with PCD, which is by far the most used method and the de facto standard. We make this choice in an effort to be both consistent and clear in our writing and experimentation rather than to be completely exhaustive in our choice of comparisons.

- We believe that, on top of providing a new objective function, the extended mean-field approach possesses two distinct advantages. First, it allows for systematic improvement by adding higher-order terms (e.g. [16]). Second, it can be directly generalized to semi-RBMs (which possess additional of visible-to-visible couplings) or even to stacked RBMs. We leave these extensions for future work, as we first intend to demonstrate that the extended mean-field approach can perform well on a pure RBM. However, nothing in the approach relies on a bipartite, or single layer, structure in particular.

- We strongly believe that the usefulness of generative models, such as RBMs, is not restricted to their use as feature selectors for deep nets, but that they can also be used as structured priors for inference on a wide range of graphical models, as well as being a very interesting analytical topic of study for learning problems in general. To quote Geoffrey Hinton (see https://www.reddit.com/r/MachineLearning/comments/2lmo0l/ama_geoffrey_hinton) "[...] most people have lost interest in generative models. But I am sure they will make a comeback in a few years and I think most of the pioneers of deep learning agree." In this respect, using RBMs in conjunction with efficient statistical inference methods (for instance, Approximate Message Passing as in [arXiv:1502.06470]) is a promising direction for tasks as diverse as denoising, factorization, matrix completion, and structured compressed sensing, to name a few.

** Response to Reviewer 5:

- See response to Reviewer 4.